# Position: RL Should Be Used to Adjust Foundation Models, NOT Abused

**Ting Huang** [1] [*]  **Zeyu Zhang** [1] [*] [†]  **Hao Tang** [1] [‡]

## Abstract

This position paper argues that reinforcement learning (RL) should be used to *adjust* foundation models after pretraining and cold-start supervision, not *abused* as a default recipe for capability creation or early-stage training. We view RL as a high-cost, high-leverage post-training operator that most reliably reallocates probability mass toward behaviors a model can already express, improving correctness, consistency, and constraint satisfaction. This is not an impossibility claim: RL may discover new behaviors when meaningful supervision or scaffolding is unavailable and verification is strong, but under current foundation-model practice it should not be treated as the default path to reasoning capability. Across modalities and domains, we emphasize a recurring regularity: supervision establishes usable reasoning structure, whereas RL mainly sharpens behavior under constraints. We further advocate reward minimalism through auditable, verifiable, and minimally composed rewards, and discuss self-supervised RL only as a boundary case for structured interaction settings. Together, these arguments motivate treating RL as a disciplined adjustment stage with explicit entry criteria and compute-accountable evaluation. Blog: https://aigeeksgroup.github.io/DontAbuseRL/.

## 1. Introduction

Post-training has become the dominant lever for turning large pretrained foundation models into systems that are reliably helpful, safe, and task-effective. Across language and multimodal models, reinforcement learning (RL) is increasingly treated as the final "capability and alignment dial" after pretraining and supervised fine-tuning (SFT), through

---
[*]Equal contribution. [†]Project lead. [1]School of Computer Science, Peking University. Correspondence to: Hao Tang <bjdxtanghao@gmail.com>.

*Proceedings of the 43rd International Conference on Machine Learning*, Seoul, South Korea. PMLR 306, 2026. Copyright 2026 by the author(s).

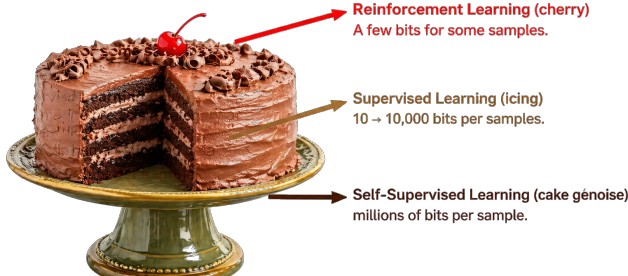

*Figure 1.* **Yann LeCun's Cake Analogy.** A conceptual illustration comparing learning paradigms: self-supervised learning as the cake (main bulk of information), supervised learning as the icing, and reinforcement learning as the cherry on top.

preference optimization, constraint satisfaction, and verifiable reward signals (Ouyang et al., 2022; Schulman et al., 2017; Bai et al., 2022; Ziegler et al., 2019). This staged view is consistent with the intuition illustrated in Figure 1: most reusable information is acquired before RL, while RL is most naturally used as a small but high-leverage adjustment layer on top of pretrained and supervised structure. This shift is pragmatic because many deployment-critical behaviors, such as instruction adherence, refusal policies, tool-use discipline, and long-horizon consistency, are difficult to specify as supervised targets but can be expressed as feedback or outcome-based objectives.

At the same time, RL is costly and failure-prone at the foundation-model scale. Compared with supervised objectives, RL typically demands repeated sampling, near on-policy updates, stabilization heuristics, and tight coupling between prompting formats and reward definitions (Schulman et al., 2017; Rafailov et al., 2023). Small reward design choices can induce large behavioral shifts, including reward hacking, style regressions, and brittleness under distribution shift. As a result, RL is powerful but easy to misuse: it is tempting to treat RL as a universal solution, apply it early, and iterate until the desired behavior appears.

This has contributed to a growing confusion in recent discourse. RL is sometimes framed as the mechanism that creates reasoning ability, rather than a mechanism that selects, amplifies, and stabilizes reasoning behaviors already latent after pretraining and structured supervision (Christiano et al., 2017; Ouyang et al., 2022). The renewed popularity of "RL-zero" narratives strengthens this impression by suggesting that reward optimization alone can serve as a primary route to reasoning competence (Guo et al., 2025).

Yet controlled comparisons increasingly point to a different regularity: cold-start supervision often establishes usable reasoning structure, while RL most consistently improves correctness, consistency, and constraint satisfaction as targeted refinement (Wei et al., 2025). Put differently, RL often reshapes how a model uses its capabilities, not whether those capabilities exist.

**We argue that RL should be prioritized as a post-training adjustment operator because it is most reliable, controllable, and compute-effective at refining behaviors that a model can already express, and it should not be abused as a default recipe for capability creation or early-stage training.** This is a default-practice claim, not an impossibility claim. RL may discover new behaviors when meaningful supervision or scaffolding is unavailable, verification is strong, and interaction is necessary for discovery. However, under current foundation-model practice, such RL-first regimes should be treated as exceptions requiring stronger evidence and compute-accountable reporting, rather than as the default path to reasoning capability.

Our position has four practical implications. First, staged training with cold-start SFT followed by RL refinement should be the default for complex behaviors unless strong evidence justifies RL-first pipelines (Wei et al., 2025; Guo et al., 2025). Second, RL claims should disentangle RL effects from representational and structural scaffolding provided by pretraining and supervision. Third, reward design should prioritize auditability and minimal composition: auditable, verifiable rewards can reduce dependence on opaque or over-engineered reward stacks (Zhou et al., 2025). Finally, we discuss self-supervised RL only as a boundary case for structured interaction settings, where self-generated experience is useful only when grounded in verifiable signals and designed to resist self-reinforcing errors (Fan et al., 2025; Liu et al., 2025c).

This question matters because if "more RL" becomes synonymous with "more capability," the field may waste compute, reduce reproducibility, and obscure which ingredients build reasoning structure versus which merely polish it.

**Roadmap.** We first clarify what RL changes and what it does not in modern post-training, then argue for adjustment-oriented principles with lightweight observations, steelman two alternatives, and conclude with decision criteria, implications, and a call to action.

**Conflict of Interest Disclosure.** The authors declare no financial conflicts of interest related to this work.

## 2. Context and Prior Work

RL-based post-training sits at the intersection of preference-based alignment, scalable supervision, and reasoning-oriented post-training. The relevant question for this position paper is not whether RL *can* improve behavior, but *when* it is the right lever, what it changes, and what evidence is needed to distinguish capability creation from capability adjustment. We summarize dominant practices and recurring misinterpretations that motivate treating RL as adjustment by default, while allowing RL-first as an exception when meaningful supervision or scaffolding is genuinely unavailable.

### 2.1. What the Field Currently Does

**RLHF and preference optimization.** Modern alignment pipelines popularized a staged recipe in which supervised instruction tuning is followed by RL on learned rewards, commonly implemented with policy-gradient methods such as PPO (Schulman et al., 2017; Ouyang et al., 2022). To scale feedback, systems often replace humans with AI feedback or rule-based principles, yielding RLAIF-style and constitutional approaches (Bai et al., 2022; Lee et al., 2023). In parallel, preference-optimization objectives such as DPO (Rafailov et al., 2023), ORPO (Hong et al., 2024), and KTO (Ethayarajh et al., 2024) reduce the operational burden of on-policy RL by optimizing policies directly from preference pairs.

**Reward modeling, verifiers, and proxy optimization.** Reward modeling is often a bottleneck in RLHF, motivating benchmarks that evaluate subtle preference distinctions, style sensitivity, and out-of-distribution behavior (Lambert et al., 2025). In multimodal settings, RL-based alignment has been used to reduce hallucinations and improve factual grounding via fine-grained feedback or fact-augmented rewards (Yu et al., 2024; Sun et al., 2024). These efforts show that RL can produce tangible behavioral improvements while also foregrounding reward generalization, verifier robustness, and proxy optimization as recurring limiting factors. At scale, RL further couples training to sampling and reward fidelity; imperfect objectives invite specification gaming and reward hacking, and practical pipelines often rely on KL regularization, reference policies, and other stabilizers to control drift (Skalse et al., 2022; Korbak et al., 2022).

**Reasoning-oriented and resource-aware RL pipelines.** Recent reasoning-oriented systems increasingly use RL after structured supervision rather than as an unscaffolded starting point. For example, MoRL (Wang et al., 2026) combines supervised fine-tuning with verifiable-reward RL for motion reasoning, while OralGPT-Plus (Fan et al., 2026) uses expert-curated diagnostic trajectories before reinspection-driven RL for medical visual reasoning. A related resource-aware line, such as RARRL (Liu et al., 2026), learns when to invoke costly LLM reasoning under latency and budget constraints. Together, these works reinforce our framing:

RL is valuable for refining or orchestrating costly behaviors under explicit constraints, but its scaffolds, costs, and decision boundaries should be reported rather than hidden.

## 2.2. What We Systematically Misinterpret

**Misinterpretation 1: "RL creates reasoning."** A recurring narrative frames RL as the ingredient that *produces* reasoning, rather than a mechanism that *selects, amplifies, and stabilizes* behaviors already latent after pretraining and structured supervision. This framing is reinforced by RL-first and "RL-zero" storytelling around reasoning-oriented models, even when the training stack includes scaffolding such as prompt formats, curated data filters, expert trajectories, or implicit supervision (Guo et al., 2025). In contrast, controlled comparisons in multimodal reasoning suggest a different regularity: cold-start supervision often establishes usable reasoning structure, while RL more consistently sharpens correctness and constraint satisfaction as refinement (Wei et al., 2025). Recent cross-domain systems in motion and medical visual reasoning follow the same staged pattern (Wang et al., 2026; Fan et al., 2026). Our claim is therefore not that RL can never discover new behaviors, but that claims of reasoning creation require stronger evidence than claims of reasoning elicitation, selection, or refinement.

**Misinterpretation 2: "More sophisticated rewards are necessarily better."** When reward hacking appears, the default response is often to increase reward complexity by enlarging reward models, adding features, mixing objectives, or applying elaborate shaping. Yet richer proxies can still be exploited and may simply shift the failure surface rather than eliminate it (Skalse et al., 2022). Reward-model benchmarks likewise show that the central difficulty is not only model capacity, but also coverage, calibration, and evaluation under shift (Lambert et al., 2025). This motivates a different principle: auditable, verifiable, and minimally composed signals can be preferable to opaque reward complexity when sufficient for the target behavior. Here, auditability is practitioner-facing: reward components should be inspectable, stress-testable, and attributable during failure analysis, even if the learner optimizes an aggregated scalar reward.

**Misinterpretation 3: "Preference objectives remove the need for stage discipline."** DPO-style objectives reduce on-policy complexity, but they do not remove the need for cold-start structure or careful data construction. Preference-optimization works still note that supervised fine-tuning helps convergence and defines the policy behavior manifold before preference optimization reshapes it (Rafailov et al., 2023; Hong et al., 2024). Treating preference optimization as a universal substitute for staged training risks reproducing RLHF failure modes in another form, including sensitivity to data composition and proxy misalignment.

**Under-discussed: compute accounting and evaluation incentives.** As reward-model benchmarks and alignment leaderboards improve, systems can overfit to what is measured while under-reporting sampling budgets, annotation costs, verifier failures, and stability interventions. This makes RL-first pipelines difficult to compare with staged alternatives in a compute-accountable way (Lambert et al., 2025). Resource-aware orchestration further shows that reasoning itself has latency and budget costs that should be explicitly modeled rather than assumed away (Liu et al., 2026). This motivates explicit entry criteria and compute-aware evaluation of RL stages.

## 2.3. Why the Current Trajectory Is Inadequate

The current trajectory trends toward normalizing RL-first pipelines for reasoning and alignment because they can yield impressive headline behaviors. However, this direction is inadequate as a *default* trajectory, even if it remains justified in exception regimes where meaningful supervision or scaffolding is unavailable and verification is strong.

First, it conflates *behavioral sharpening* with *capability creation*. When RL is credited for reasoning emergence without disentangling pretraining, cold-start supervision, expert trajectories, or structured prompting, the field risks drawing incorrect scientific conclusions and propagating unnecessarily costly recipes (Wei et al., 2025; Guo et al., 2025; Wang et al., 2026; Fan et al., 2026). Second, it scales the most failure-prone component of the stack: optimizing imperfect objectives structurally invites reward hacking and proxy exploitation (Skalse et al., 2022). Third, it pushes the field toward less reproducible and less compute-transparent research, since sampling details, reward choices, verifier robustness, reasoning latency, and stabilizers are often under-specified (Schulman et al., 2017; Liu et al., 2026). These issues motivate treating RL as a disciplined adjustment stage with clear entry criteria and auditable objectives, rather than as a default early-stage recipe.

# 3. Main Argument: Use RL for Adjustment

This section defends our central claim through adjustment-oriented principles. RL should close residual gaps after supervision or scaffolding when available, while RL-first should be reserved for regimes where such signals are unavailable and verification is strong.

## 3.1. RL for Adjusting Foundation Models

> **Pillar 1:** RL should be used to adjust foundation models after pretraining, particularly when a model remains unsatisfactory in certain aspects.

Across recent "R1-style" post-training pipelines, RL most commonly appears as a *refinement step* that pushes a partially capable model toward higher reliability under explicit constraints. Examples across 3D understanding (Huang et al., 2025b), embodied navigation (Liu et al., 2025a), VLA (Ye et al., 2025), motion generation (Ouyang et al., 2026), mobile robotics (Huang et al., 2025a), and world models (Feng et al., 2025) follow a similar staged pattern: supervised or structured traces first establish the reasoning/action format, and RL then improves correctness, constraint satisfaction, or long-horizon stability. These examples point to a common entry condition: *apply RL when the model can already produce the desired behavior sometimes, but fails too often, violates constraints, or is inconsistent.* In that regime, RL acts as a selector and stabilizer, shifting probability mass toward correct and policy-compliant trajectories.

### 3.2. RL Should Not Be Abused

> **Pillar 2:** When meaningful supervised or scaffolded signals are available, they should usually precede RL.

The first-order question is not simply whether RL is expensive, but whether meaningful supervised or scaffolded signal is available. When such signal exists at reasonable fidelity, it should usually establish a stable behavior manifold before RL is invoked. RL cost, instability, and reporting burden remain important, but they are secondary to this more basic decision criterion.

The core cost of RL is structural: optimization is coupled to sampling and reward fidelity, making progress sensitive to prompt formats, reward definitions, and stabilization heuristics. When RL is used before the behavior space has been shaped by pretraining, supervision, or other scaffolds, much of the budget may be spent exploring unstable trajectories rather than refining a usable behavior manifold. Controlled comparisons show that staged SFT-then-RL pipelines are more stable than RL-only or SFT-only settings, with RL success strongly dependent on cold-start quality (Wei et al., 2025). Thus, RL-first is not impossible, but carries a higher burden of proof: authors should justify why meaningful supervision is unavailable, why exploration is necessary for discovery, and why cheaper alternatives such as improving data, prompting, programmatic supervision, or model specification are insufficient.

### 3.3. RL Should Not Be Casually Credited with Creating Reasoning

> **Pillar 3:** RL should not be assumed to create reasoning from scratch when pretrained, supervised, or scaffolded structure already explains the gains.

A practical way to interpret Pillar 3 is to separate *representation* from *selection*. In many current foundation-model pipelines, pretraining and cold-start supervision provide the representational and procedural scaffolding that makes step-by-step reasoning elicitable. RL then reallocates probability mass toward trajectories that satisfy constraints and pass checks, improving consistency and reducing error modes that supervised training alone may not eliminate (Wei et al., 2025). Thus, "RL improves reasoning" is plausible if it means higher correctness on tasks the model could already sometimes solve, better constraint satisfaction, or more stable multi-step traces. By contrast, "RL creates reasoning from scratch" is a stronger causal claim that should control for pretrained structure, prompt scaffolds, data filters, and supervised traces, and should report sampling budgets, verifier reliability, and stability interventions. Our argument is not that RL can never discover new behaviors, but that reasoning-creation claims require stronger evidence than reasoning-refinement claims.

### 3.4. Reward Functions Should Be Auditable and Minimally Composed

> **Pillar 4:** Reward design should prioritize auditability, verifiability, and minimal composition over opaque reward complexity.

RLVR replaces opaque preference proxies with rewards checked by verifiers, tests, or deterministic rules. It improves auditability because the signal is derived from externally checkable criteria rather than opaque learned reward models. Here, auditability is practitioner-facing: reward components should be testable, stress-testable, and attributable during failure analysis, even if the learner optimizes only an aggregated scalar reward.

Recent analyses show that RLVR can improve reasoning with clearer objectives, but verifiers can still be noisy or exploitable (Wen et al., 2025; Cai et al., 2025; Tu et al., 2025). Thus, minimal composition is useful not because reward design is easy, but because auditable rewards are easier to diagnose and stress-test. In R1-style systems, verifiable rewards such as alignment, consistency, formatting, and task-grounded correctness are easier to debug than a monolithic learned notion of "good reasoning" (Ye et al., 2025; Huang et al., 2025b; Liu et al., 2025a).

### 3.5. Boundary Case: SSRL under Structured Interaction

> **Boundary Principle:** SSRL is relevant only in structured interaction regimes where meaningful supervision is weak, verification is strong, and self-generated experience can be externally audited.

Self-supervised reinforcement learning is not a generic replacement for pretraining or supervised scaffolding. We discuss it only as a boundary case: when external supervision is weak, the environment supplies verifiable progress signals,

and interaction itself is the substrate for discovering useful behavior. Classical self-supervised RL shows that intrinsic objectives can drive exploration and downstream competence without task labels (Sekar et al., 2020). Recent systems such as EvoVLA (Liu et al., 2025c), Spatial-SSRL (Liu et al., 2025b), and Web World Models (Feng et al., 2025) illustrate this limited regime, where self-generated experience can help when paired with memory, structured environments, and verifiable progress signals. Under our framework, these examples do not overturn stage discipline; they specify when interaction-based learning may substitute for missing supervision. Accordingly, authors should report signal stability, held-out generalization, and failure modes of self-generated objectives, rather than only short-term proxy gains.

### 3.6. Risks of Inaction

If the research community defaults to RL-first recipes for capability creation, three risks follow. First, incentives may drift toward compute-heavy pipelines whose success depends on sampling budgets and fragile stabilizers. Second, causal attribution may degrade, with gains credited to RL even when they primarily depend on cold-start structure, data curation, prompt formats, or other scaffolds (Wei et al., 2025). Third, safety and governance risks grow when optimization targets are opaque, since reward hacking and specification gaming are structural possibilities in proxy optimization. Treating RL as disciplined adjustment, with auditable rewards and explicit entry criteria, mitigates these risks without denying that RL-first may be justified when meaningful supervision is unavailable and verification is strong.

## 4. Case Study and Empirical Observation

This section provides lightweight, illustrative evidence for viewing RL as a post-training adjustment operator rather than a default early-stage capability-creation recipe. We use these observations to motivate the position, not to present benchmark-style performance claims, and to clarify what stronger evidence would be needed for RL-first capability-creation claims.

### 4.1. Observation 1: RL Gains Track Cold-Start Quality

A recurrent pattern across recent post-training reports is that RL gains depend strongly on cold-start quality. In multimodal reasoning, controlled comparisons show that staged training, i.e., cold-start SFT followed by RL, yields more stable improvements than RL-only or SFT-only settings, and that RL success correlates with supervised initialization strength (Wei et al., 2025). This is consistent with the adjustment view: if RL mainly reallocates probability mass toward better trajectories, the policy must already assign non-trivial probability to such trajectories.

**Why this supports our position.** If RL were generally a compute-efficient default mechanism for *creating* reasoning from scratch, RL-only pipelines should dominate staged pipelines once sufficient compute is spent. The observed dependence on cold-start quality instead suggests that RL is most effective when sharpening behaviors that are already partially present. Thus, RL-first reasoning claims require stronger evidence whenever meaningful supervised or scaffolded signals are available.

**A minimal check.** A lightweight test is to fix the RL budget and vary only cold-start strength, e.g., by subsampling supervised traces or removing structured formatting, while holding prompt format, verifier, sampling budget, and stabilizers constant. If RL gains collapse as cold-start quality drops, the result supports the adjustment interpretation. Conversely, if RL remains stable and improves under weak or absent scaffolding, the result would provide evidence for the stronger RL-first interpretation.

### 4.2. Observation 2: Verifiable Rewards Improve Reliability, but Reveal a "Verifier Bottleneck"

A second pattern is that RL with verifiable signals improves reliability when the check is clear, but becomes limited by verifier coverage and robustness. Many R1-style pipelines show that checkable signals, such as format validity, consistency constraints, and executable outcomes, can produce meaningful improvements without a learned reward model (Wen et al., 2025). However, imperfect verifiers can be noisy or exploitable, shifting optimization from "solve the task" to "satisfy the verifier" (Cai et al., 2025; Tu et al., 2025).

**Why this supports our position.** This supports reward minimalism as auditability, verifiability, and minimal composition, not oversimplification or the claim that reward design is easy. When verifiers become the bottleneck, the remedy is not necessarily a larger reward model, but better verifier evaluation, adversarial stress testing, and transparent reporting of failure modes.

**Connection to staged training.** Verifier bottlenecks also explain why cold-start scaffolds remain useful even when rewards are verifiable. A supervised or scaffolded policy narrows the search space before RL, making it less likely that optimization exploits verifier loopholes rather than task-relevant behavior. Thus, verifiable rewards and stage discipline are complementary rather than competing design choices.

**A minimal diagnostic.** A useful diagnostic is to compare verifier behavior before and after RL: report pass rates for each check, dominant failure categories, and sensitivity to small verifier perturbations. If policy gains disappear under minor changes to verifier wording, thresholds, or formatting

rules, the result should be interpreted as reward overfitting rather than robust behavioral improvement. This diagnostic also helps separate genuine task improvement from verifier overfitting, making reward failures attributable to concrete components rather than to an opaque scalar score.

### 4.3. Observation 3: "R1-style" Cross-Domain Pipelines Use RL as Refinement, Not as First Principle

A third empirical signal comes from cross-domain convergence. Across 3D understanding (Huang et al., 2025b), navigation (Liu et al., 2025a), VLA (Ye et al., 2025), motion generation (Ouyang et al., 2026), and mobile robotics (Huang et al., 2025a), recent R1-style pipelines repeatedly adopt a two-stage structure: cold-start supervision first establishes structured reasoning traces, and RL then improves constraint satisfaction, formatting reliability, or long-horizon stability. In world-model settings, Web World Models (Feng et al., 2025) similarly emphasize structured environments where objectives can be audited, making RL an adjustment mechanism over an explicit substrate rather than an opaque preference-optimization loop.

**Why this supports our position.** This cross-domain consistency is informative even without a new benchmark: independent systems repeatedly converge on the same engineering equilibrium. If RL were the most efficient default path to capability creation, RL-first pipelines should dominate more consistently in domains where supervised or scaffolded signals are available. Instead, the observed pattern is that supervision or structure defines the behavior manifold, while RL improves reliability within that manifold.

**A minimal check.** A lightweight test is to hold the RL budget fixed and weaken the scaffold, e.g., by removing structured formatting, subsampling supervised traces, or replacing expert trajectories with weaker synthetic ones. If RL refinement becomes less stable or yields smaller gains in constraint satisfaction, the result supports the adjustment interpretation. Conversely, if RL remains robust under weak or absent scaffolding, it would provide evidence for the stronger RL-first interpretation.

### 4.4. What These Observations Do and Do Not Claim

These observations do not claim that RL-first pipelines can never work, nor that RL is unnecessary. They support a narrower conclusion: *when meaningful supervised or scaffolded signals are available, RL is most reliable and compute-effective as an adjustment stage once the target behavior is already expressible*. In this regime, failures are often dominated by reward design, verifier coverage, sampling cost, and stabilization details, rather than by the complete absence of capability. RL-first remains justified when meaningful supervision or scaffolding is genuinely unavailable, verification is strong and low-noise, and interaction is necessary for discovery. Such cases should be argued explicitly and evaluated with compute-accountable reporting, including sampling budgets, verifier robustness, stabilizer sensitivity, and comparisons against minimal cold-start scaffolds. This is the core stance of our adjustment-oriented framework.

## 5. Alternative Views

We present two credible alternative views and explain why we favor "RL for adjustment" under compute-accountable, falsifiable criteria. Our goal is not to rule out RL-first regimes, but to clarify when they should be treated as exceptions rather than defaults.

### 5.1. Alternative View A: RL Can Enable Reasoning-Like Behaviors

*Opposing perspective.* "RL-zero" style results argue that outcome-based RL can *enable* reasoning-like behaviors even without supervised cold-start data, potentially yielding longer chains of thought, self-correction, and strategy discovery. DeepSeek-R1-Zero (Guo et al., 2025) is often cited as emblematic evidence.

*Strengths.* This view is attractive because it promises scalable improvement without expensive reasoning traces, aligns with RL as strategy discovery under sparse outcomes, and highlights behaviors that may be difficult to specify through direct supervision alone.

*Our response.* We agree that RL can produce reasoning-like behaviors and may even discover new behaviors in some regimes. Our disagreement is with the stronger conclusion that RL should therefore be the *default* capability-creation recipe for foundation models. First, RL-zero results are also consistent with *selection and amplification*: pretrained models already contain substantial structure, and RL can shift probability mass toward trajectories that exploit it; imperfect objectives can also amplify exploitable behaviors (Skalse et al., 2022). Second, flagship RL-zero pipelines report undesirable behaviors such as repetition and readability issues, and later introduce cold-start data, suggesting that stage discipline remains useful in practice (Guo et al., 2025). Third, controlled comparisons suggest that staged training is more consistently effective when meaningful supervision is available, and that RL gains depend on cold-start quality.

**What would change our mind.** Compute-accountable evidence that RL-only reliably matches or exceeds staged pipelines under controlled cold-start removal, without instability, regressions, or hidden scaffolding, would weaken our claim. Such evidence should report sampling budgets, verifier reliability, stabilizers, and comparisons against minimal supervised or scaffolded seeds.

**Conclusion for View A.** We do not dispute the phenomenon. Rather, we argue for a narrower interpretation: RL can amplify and sometimes discover reasoning-like behaviors, but current evidence does not justify treating RL as the default path to reasoning capability whenever supervised or scaffolded signals remain available.

### 5.2. Alternative View B: RL-First Is the Most Scalable Path When Supervision Is Costly

*Opposing perspective.* In some domains, the issue is not merely that high-quality supervision is costly, but that meaningful supervised targets are unavailable, incomplete, or poorly defined. Examples include theorem discovery, open-ended strategy search, self-play, and interactive environments where the target behavior must be discovered through trial, verification, or environment feedback. In such settings, RL-first appears attractive because scalable signals such as verifiable rewards, self-play, interaction, and automated feedback may be more informative than static demonstrations.

*Strengths.* This view correctly emphasizes scalability, discovery, and deployment-relevant objectives. It also identifies the strongest challenge to our position: in some settings, the absence of meaningful supervised targets is a structural limitation rather than merely an annotation-cost issue. For example, in theorem discovery or open-ended strategy search, the desired behavior may not be identifiable from static demonstrations, making interaction and verification part of the learning substrate itself.

*Our response.* We agree that these regimes are real and important. Our disagreement is with generalizing them into a default recipe for foundation-model post-training. First, the main decision criterion should be whether meaningful supervised or scaffolded signal is available, not RL cost alone. When such signal exists, even in small, synthetic, programmatic, or format-level form, it often provides a cheaper and more stable scaffold than unconstrained RL exploration. Second, RL cost still matters because sampling, verifier reliability, and stabilizers create substantial reproducibility debt. Third, RL-first claims require stronger reporting: sampling budgets, verifier acceptance and failure modes, stabilizer sensitivity, robustness to reward perturbations, and comparisons against minimal cold-start scaffolds.

**What would change our mind.** Consistent evidence that RL-first is more reliable or compute-effective than staged alternatives when meaningful supervision is genuinely unavailable and verification is strong would update our stance. Such evidence would be especially compelling if it remained robust under transparent sampling, stability, and verifier-stress reporting.

**Conclusion for View B.** RL-first can be justified when

meaningful supervision or scaffolding is unavailable, verification is strong, and interaction is necessary for discovery. As a default for foundation-model post-training, however, it remains premature because brittleness, proxy failures, and reproducibility burdens scale with reliance on RL (Skalse et al., 2022).

### 5.3. Where We Expect Genuine Disagreement to Persist

The core disagreement is not whether RL can improve behavior or sometimes discover new behavior, but what should be treated as the default causal interpretation and training recipe under current foundation-model practice. If meaningful supervision or scaffolding is available, we view RL primarily as an adjustment operator; stage discipline, auditable rewards, and compute accountability should therefore be the defaults. If RL is claimed to create capabilities, the evidence burden should be higher: authors should rule out hidden scaffolds, report sampling and stabilization costs, stress-test verifiers, and compare against minimal supervised or scaffolded seeds. This is the distinction our adjustment-oriented framework makes operational.

## 6. Practical Guidelines for RL as Adjustment

We propose a compact decision framework: treat RL as a disciplined *adjustment stage* with explicit entry criteria, auditable rewards, and compute-accountable reporting. The framework is implementation-agnostic and applies to PPO-style RLHF, GRPO-style variants, preference optimization, and RLVR, while constraining *when* RL should be used and *what claims* can be credibly attributed to it.

### 6.1. Core Idea or Architecture

**RL as an adjustment operator.** We model post-training as a staged pipeline:

Pretraining → Cold-start / Scaffolding → **RL Adjustment** → Release & monitoring.

RL is most reliable when applied *after* the target behavior is already elicitable in some conditions. In this framing, RL is not the default mechanism for capability creation; it reallocates probability mass toward correct, policy-compliant, and stable trajectories, and reduces diagnosable residual failures after pretraining, cold-start supervision, or other scaffolds. This does not rule out RL-first regimes, but treats them as exceptions requiring stronger evidence when meaningful supervised or scaffolded signals are unavailable. Here, scaffolding may include supervised traces, synthetic curricula, programmatic feedback, format constraints, tool demonstrations, or other structures that make the target behavior initially elicitable.

**Adjustment targets.** Prioritize behaviors that are deployment-critical, poorly specified by imitation alone, and checkable with auditable signals, such as constraint

satisfaction, long-horizon consistency, format discipline, safety policy adherence, tool-use correctness, and execution validity (Ouyang et al., 2022; Bai et al., 2022).

## 6.2. Key Components or Mechanisms

**Component 1: Entry criteria.** Before invoking RL, first ask whether meaningful supervised or scaffolded signal is available. If so, use it to establish a stable behavior manifold and reserve RL for residual failures. Invoke RL when the target behavior is **elicitable**, failures are **diagnosable**, and success is **auditable** via verifiers, tests, or externally checkable criteria. If RL is used before such scaffolding exists, authors should justify why meaningful supervision is unavailable, why interaction or exploration is necessary for discovery, and why cheaper alternatives such as data, prompting, programmatic supervision, or model specification are insufficient. This criterion is intentionally prior to compute accounting: if meaningful supervision exists, ignoring it weakens the causal attribution of RL gains.

**Component 2: Stage discipline with explicit exceptions.** Use cold-start supervision or other scaffolds to establish a stable behavior manifold, then apply RL for refinement. This matches evidence that RL effectiveness depends strongly on cold-start quality (Wei et al., 2025). RL-first is defensible mainly when meaningful supervision or scaffolding is genuinely scarce, verification is strong and low-noise, and the target behavior must be discovered through interaction, self-play, or environment feedback. In those regimes, RL-first should be evaluated as an exception with stronger evidence requirements, not normalized as the default recipe.

**Component 3: Reward minimalism.** Prefer auditable, verifiable, and minimally composed reward signals over opaque reward stacks. Minimalism does not mean that reward design is easy, nor that sparse rewards are always better than dense rewards. Rather, reward components should be externally inspectable, independently stress-testable, and attributable during failure analysis (Wen et al., 2025). Because verifiers can be noisy or exploitable, even simple checks should be evaluated under perturbations and adversarial cases (Cai et al., 2025; Tu et al., 2025). If learned reward models or complex proxy stacks are unavoidable, treat them as fallible proxies and evaluate them under distribution shift (Lambert et al., 2025).

**Component 4: Guardrails and compute-accountable reporting.** Imperfect objectives invite reward hacking and specification gaming (Skalse et al., 2022). Use verifier stress tests, regression suites for non-target properties such as readability and safety behavior, and budget-sensitivity checks to ensure gains are not artifacts of extreme sampling. Report total sampled trajectories, effective batch sizes, verifier pass rates, sensitivity to key stabilizers such as KL strength or sampling temperature, and detected regressions (Korbak et al., 2022). For RL-first claims, also compare against minimal cold-start or scaffolded baselines and report verifier robustness under perturbations. This makes claims about capability creation, elicitation, and refinement comparable across systems.

**Boundary component: SSRL under structured interaction.** For agents and world models, self-supervised RL is relevant when meaningful supervision is weak, interaction is necessary for discovery, and the environment supplies verifiable progress signals (Sekar et al., 2020). Recent systems emphasize interaction loops and memory that remain auditable (Feng et al., 2025; Liu et al., 2025c). Under our framework, SSRL is not a generic replacement for pretraining or supervised scaffolding; it is a boundary case that should be evaluated by held-out generalization, signal stability, and failure modes of self-generated objectives, not only short-term proxy gains.

## 6.3. Evidence-Informed Justification

This framework consolidates empirical regularities and failure analyses rather than proposing a new algorithm. Staged recipes are repeatedly reported as more reliable than RL-only when meaningful supervised signals are available, and RL gains depend strongly on cold-start quality (Wei et al., 2025). Cross-domain R1-style pipelines converge on supervised scaffolds followed by RL refinement (Huang et al., 2025b; Liu et al., 2025a; Ye et al., 2025; Ouyang et al., 2026; Huang et al., 2025a). RLVR results emphasize that verifiable signals can be effective and auditable, while verifier noise motivates stress testing and minimal composition (Wen et al., 2025; Cai et al., 2025; Tu et al., 2025). Structured environments support SSRL only when they provide signals aligned with success and limit self-reinforcing errors (Sekar et al., 2020; Feng et al., 2025).

**Summary.** Use RL as a high-leverage adjustment stage after cold-start alignment or scaffolding, prefer auditable and verifiable rewards, and report compute so that claims are reproducible and comparable. When RL-first is used, state why meaningful supervision is unavailable and provide stronger evidence through sampling budgets, verifier stress tests, stabilizer disclosure, and comparisons against minimal scaffolded baselines.

# 7. Call to Action

To make RL post-training scientifically clearer and operationally safer, the community should treat RL as an *adjustment stage* and hold RL-based claims to higher standards of auditability, falsifiability, and compute transparency. RL-first should not be dismissed categorically, but it should be argued explicitly when meaningful supervision or scaffolding is unavailable and verification is strong.

### 7.1. What Researchers Should Do

**Use stage discipline by default.** Before invoking RL, report whether meaningful supervised or scaffolded signal is available. When it is available, use it to establish the behavior manifold and report how RL gains change with cold-start strength (Wei et al., 2025). RL-first claims should match total compute, disclose stabilizers, and compare against minimal supervised or scaffolded baselines.

**State falsifiable claims.** Separate supervision scaffolds from RL effects; specify the missing behavior, the check that defines success, and whether the claim concerns capability creation, elicitation, or refinement. Claims that RL creates new reasoning behaviors should rule out alternative explanations such as pretrained structure, prompt scaffolds, data filters, and supervised traces.

**Report rewards and costs transparently.** Use externally checkable objectives when possible, and report verifier pass rates, failure categories, robustness under perturbations, sampled trajectories, effective batch sizes, stabilizers, verifier calls, and detected regressions (Wen et al., 2025; Cai et al., 2025; Tu et al., 2025). Reward minimalism should mean auditability, verifiability, and minimal composition, not the assumption that reward design is easy.

### 7.2. What Institutions or Companies Should Do

**Institutionalize transparency.** Adopt release and publication checklists that require reporting sampling cost, reward definitions, verifier behavior, and stabilizers, especially when RL is claimed to replace supervision or create new capabilities. Such checklists should also ask whether the reported gains depend on hidden scaffolds, unreported filtering, or unusually large sampling budgets.

**Invest in verification and monitoring.** Fund stress-tested verifiers, adversarial evaluation, and regression suites for non-target behaviors. Pair RL adjustment with monitoring and rollback processes so that reward-induced regressions can be detected after deployment.

### 7.3. What the ML Community Should Prioritize

**Shared standards.** Develop norms that distinguish capability creation, capability elicitation, and behavioral refinement. The community should prioritize benchmarks for verifier robustness and proxy gaming, reporting standards for RL sampling and stabilizers, and comparisons between RL-first pipelines and minimal scaffolded alternatives. These standards would make RL progress easier to reproduce, audit, and compare across labs.

**What not to claim.** Post-training gains should not be credited to RL alone unless pretraining scale, prompt format, supervised traces, data filtering, verifier design, and sampling budget have been controlled. A clearer reporting norm is to specify whether RL created a new behavior, elicited a latent one, or refined an already observable one.

## 8. Broader Implications

Reframing RL as an adjustment stage improves scientific clarity by separating pretraining, supervision, and other scaffolds from RL's trajectory-selection effects. It distinguishes capability creation, capability elicitation, and behavioral refinement, so RL gains are not automatically attributed to new reasoning abilities. This framing does not categorically reject RL-first regimes: when meaningful supervision or scaffolding is unavailable and verification is strong, RL-first may be appropriate, but should carry stronger evidence and reporting requirements. It also promotes compute-aware practice by making sampling cost, verifier calls, stabilization details, and regression risks part of the evidence, while shifting governance toward auditable rewards, verifier robustness, stress testing, and transparent failure reporting. Cross-domain, it treats SSRL as a boundary case for structured environments where self-generated experience can be externally audited, and culturally shifts incentives from "more RL" to reproducible, auditable, and comparable reporting across labs.

## 9. Conclusion

**We argue that reinforcement learning should adjust foundation models after pretraining and cold-start alignment, not serve as a default recipe for early-stage capability creation.** This is not an impossibility claim: RL may discover new behaviors when meaningful supervision or scaffolding is unavailable, verification is strong, and interaction is necessary. Under current foundation-model practice, however, RL is most reliable as a high-cost operator that improves correctness, consistency, and constraint satisfaction by shifting probability mass toward behaviors a model can already express. The decision rule is simple: use meaningful supervision or scaffolding when available, then apply RL to close residual gaps with auditable rewards and compute-accountable reporting. The risk is normalizing "more RL" without distinguishing capability creation, elicitation, and refinement. This framing preserves RL-first exception regimes while making their evidence burden explicit: RL should receive causal credit for new capabilities only when hidden scaffolds, verifier artifacts, and sampling advantages have been ruled out. It therefore yields clearer causal claims, better reproducibility, and more governable post-training.

## Acknowledgements

This work was supported by the Fundamental Research Funds for the Central Universities, Peking University.

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
