# OpenReview forum: "Position: RL Should Be Used to Adjust Foundation Models, NOT Abused"
_ICML.cc/2026/Position_Paper_Track — ICML 2026 Position Paper Track regular_

### Official Review · Reviewer_WnFr · 2026-03-12

**Significance:** 3
**Argument Clarity:** 3
**Rating:** 5
**Confidence:** 4

**Questions:**

If you agree with the above weakness, can you reflect “4.4. What These Observations Do (and Do Not) Claim” view in the abstract as well as introduction.

**Alternative Views Section:**

Yes

**Compliance With Llm Reviewing Policy A Conservative:**

Affirmed.

**Discussion Potential:**

3

**Paper Summary:**

This position paper argues that RL compute should be used for post-tasks adaptation/finetuning: improving correctness, consistency and constraint satisfaction by shifting probability mass toward behaviours a model can already express due to pre-training and cold-start/supervised-finetuning. They argue that RL is not compute efficient, hence not advisable to use for pre-training or even without cold-start. Further, they advocate the simple reward function to prevent the reward hacking issue by the model. However, they agree that RL can improve reasoning capability by itself as well as can learn the model from scratch if we ignore the computation requirement.

**Position:**

Yes

**Position In Title:**

Yes

**Related Work:**

3

**Strengths And Weaknesses:**

The main strengths of the paper is advocating for report RL compute, disclose stabilizers techniques, and specify the missing behaviors and the check that defines success so that we can learn from each other and progress faster.

The main weakness of the paper is even though the authors agree that reinforcement learning algorithms can be used from the early stage if we ignore the compute and robust reward (function) available but the tone of the paper especially in the abstract, and introduction sounds opposite, which can mislead the community.

**Support:**

3

---

> ### Author Rebuttal · Authors · 2026-03-27
>
> We thank the reviewer **WnFr** for the positive assessment and for identifying an important presentation issue. We are especially encouraged that the reviewer sees value in the paper’s concrete recommendations around reporting RL compute, disclosing stabilizers, and specifying missing behaviors and success checks. We agree that the paper’s intended scope is stated most precisely in **Section 4.4 (“What These Observations Do (and Do Not) Claim”)**, and that this precision is not yet fully reflected in parts of the abstract and introduction.
>
> More specifically, our intention is not to argue that RL can never improve reasoning on its own, or that RL-first is unjustified in all settings. Rather, our claim is that under practical considerations of **compute efficiency, robustness, and reward reliability**, RL should not be treated as the default recipe for capability creation. The intended message is therefore narrower and more methodological: RL is best viewed, by default, as a high-cost but useful operator for **behavioral adjustment, refinement, and stabilization**, while stronger claims about capability creation or early-stage RL should require stronger evidence and clearer accounting.
>
> We appreciate the reviewer’s suggestion because it highlights a real risk of misinterpretation. **For a position paper, this matters especially: if the boundary conditions are stated only late in the paper, readers may attribute to the paper a stronger claim than the one we actually defend.** In the revision, we will explicitly bring the **Section 4.4 boundary conditions forward** into both the abstract and introduction, so that the paper’s scope is calibrated from the outset rather than only clarified later in the manuscript. Concretely, we will revise the abstract to state that the paper is **not making an impossibility claim**, but a claim about **default practice under realistic constraints.** We will likewise revise the introduction to explicitly note that RL-first may be justified in exceptional regimes—e.g., when supervision is scarce, verification is strong and low-noise, and interaction is the primary route to discovering the target behavior.
>
> We believe this change will make the paper less vulnerable to over-reading while preserving its core position. In other words, the revision will not so much soften the argument as **align the framing in the abstract and introduction with the more precise statement already given in Section 4.4.** We thank the reviewer again for pointing out this framing inconsistency so clearly; addressing it will make the paper’s contribution both clearer and harder to misread.

---

> > ### Author Rebuttal · Reviewer_WnFr · 2026-04-03
> >
> > Thanks, I will keep my positive rating.

---

### Official Review · Reviewer_wCam · 2026-03-12

**Significance:** 2
**Argument Clarity:** 2
**Rating:** 3
**Confidence:** 3

**Questions:**

See above.

**Alternative Views Section:**

Yes

**Compliance With Llm Reviewing Policy A Conservative:**

Affirmed.

**Discussion Potential:**

3

**Final Justification:**

The authors have addressed my questions. However I believe there would be major changes to the manuscript in order to reflect all changes discussed in the rebuttal.

**Paper Summary:**

The paper argues that reinforcement learning (RL) should be not be used for every training phase, but instead only used after obtaining a pretrained through cold-start supervision.
The paper outlines multiple misinterpretation/issues on using RL throughout and provide empirical observations:
1. RL redistributes the probability mass to sharpen on correct/consistent outputs, but not discovering capabilities from scratch.
2. Sophisticated reward models are better.
3. Preference objectives remove staged learning.
4. RL is less reproducible and less compute-transparent compared to supervised learning.
The paper provides two alternative views on RL enables reasoning and RL is most scalable when supervision is costly.
Finally, the paper discusses possible directions such as using criteria to determine when to apply RL, defaulting to supervision, minimal reward, guardrails, compute metric, and self-supervised RL.

**Position:**

Yes

**Position In Title:**

Yes

**Related Work:**

3

**Strengths And Weaknesses:**

## Strengths
- The paper provides a setting, specifically in cross-domain settings, that RL should not always be applied throughout training. This is corroborated by the fact that existing works obtain better results with cold-start supervision.
- The paper provides cases where learned reward models misgeneralize and RL can reward hack them.

## Comments
- While I agree that RL is computationally expensive, there are scenarios where RL can discover new behaviours that outperform supervision data, e.g. AlphaZero line of work and OpenAI Five. It is not obvious that RL cannot elicit reasoning just from training from a well-defined reward function. Secondly, the evidence in misinterpretation 1 only supports the fact that these methods do apply cold-start supervision, but not in anyway rejecting the possibility that RL can elicit reasoning.
- The term "simple reward" is very vague---for example a linear reward model is "simple" compared to a hand-designed decision-tree reward model, but either can suffer from misgeneralization. Secondly, it is not clear that how simple correlates with sparsity here---does "sophisticated" reward give denser feedback than "simple" reward? These should be clearly written out. The reward-shaping literature also suggests that hand-designing reward function is not necessarily easy [1].
- While I agree that RL algorithms tend to have many implementation details [2], this does not necessarily imply that RL is not suitable for training models from scratch, but rather it should be a call to action for researchers to come to a consensus on the "consistent" implementation.
- I fail to understand how the self-supervised reinforcement learning (SSRL) in this position paper fits into the narrative. Is the paper suggesting that we use SSRL for better exploration, in addition to the reward optimization objective, or is it suggesting that SSRL should be used for pretraining?
- I find the response to alternative view B to be missing one aspect. In many scenarios generating supervision data is significantly more difficult or even infeasible, e.g. discovering new theorems. In those cases the computation budget does not matter as the optimal model is not identifiable from the training supervision.

## References
- [1]: Knox, W. B., Allievi, A., Banzhaf, H., Schmitt, F., & Stone, P. (2023). Reward (mis) design for autonomous driving. Artificial Intelligence, 316, 103829.
- [2]: Huang, S., Dossa, R. F. J., Raffin, A., Kanervisto, A., & Wang, W. (2022). The 37 implementation details of proximal policy optimization. The ICLR Blog Track 2023.

**Support:**

2

---

> ### Author Rebuttal · Authors · 2026-03-27
>
> We thank the reviewer **wCam** for the careful and technically substantive comments. The reviewer helpfully identifies several places where the current manuscript can be read too broadly, and the most useful response is to sharpen those boundary conditions explicitly: **(i)** what exactly our claim about RL and reasoning does and does not imply, **(ii)** what we mean by “simple” rewards, **(iii)** why implementation complexity supports stronger reporting rather than a blanket rejection of RL-from-scratch, and **(iv)** how SSRL fits into the overall narrative.
>
> First, our paper is not intended to claim that RL can never discover new behaviors, nor that AlphaZero/OpenAI-Five-style settings are irrelevant counterexamples. In fact, these are exactly the kinds of regimes our framework is meant to allow as **exceptions**: supervision is genuinely scarce, interaction is the main route to discovering the target behavior, and the environment provides strong outcome feedback. Our intended claim is narrower: such examples do not by themselves justify treating RL as the default capability-creation recipe for foundation-model post-training. The disagreement is therefore less about whether RL can ever elicit or discover new behaviors, and more about what should count as the **default interpretation** and **default recipe** under realistic post-training constraints. We will revise the manuscript to make this distinction more explicit, especially by bringing the Section 4.4 boundary conditions and the Alternative View B exception regime into earlier parts of the paper.
>
> Second, we agree that the current wording around “**simple reward**” is too coarse and can be misread. Our intended meaning is not “linear is always better than nonlinear,” nor “sparse is always better than dense,” nor “hand-designed rewards are easy.” What we mean is closer to **reward minimalism and auditability**: when possible, prefer reward components that are independently checkable, minimally composed, and easier to stress-test, rather than relying by default on large opaque reward stacks. In that sense, the contrast is really between **auditable/verifiable signals** and **highly compositional opaque proxies**, not between “simple” and “sophisticated” in a purely functional-approximation sense. We will revise Pillar 4 to define this more precisely and to make clear that minimalism is not a claim that reward design is easy.
>
> Third, we agree that implementation complexity alone does not imply that RL is unsuitable for training from scratch. Our intended claim is more methodological: because RL-first pipelines couple sampling, stabilizers, and proxy fidelity, they carry a higher **reproducibility and accounting burden**, so stronger reporting is needed before such pipelines should be normalized as default practice. This is why the paper emphasizes compute-accountable reporting and stabilizer disclosure. We will revise the wording so that “implementation complexity” is presented as a reason for **stronger evidence standards**, not as a standalone impossibility argument.
>
> Fourth, the reviewer is right that the role of SSRL should be clarified. The paper is not suggesting SSRL as a generic replacement for pretraining or supervised scaffolding. Rather, SSRL is included as a **structured-interaction special case**: in world-model or agent settings where intrinsic or environment-defined signals are verifiable and the interaction loop itself is the substrate for discovery, self-supervised RL may be a scalable path for self-evolution. We will revise the SSRL discussion to make clear that this is an exception-compatible extension of our framework, not its mainline recommendation.
>
> Finally, on the point that some domains may lack feasible supervision altogether (e.g., theorem discovery), we agree this deserves a more explicit treatment in Alternative View B. These are precisely the regimes where RL-first may be most defensible. Our position is not “do not use RL there,” but rather: when supervision is genuinely unavailable, reward quality is strong, and exploration is intrinsic to the task, RL-first should be treated as a serious candidate **with stronger evidence requirements**, rather than dismissed or assumed by default. We will revise the paper to make this exception regime more explicit.
>
> > Concretely, in the revision we will:
> (i) narrow the wording of Misinterpretation 1 so it does not read as an impossibility claim;
> (ii) redefine “simple reward” more precisely in terms of **auditability, verifiability, and minimal composition**;
> (iii) clarify that implementation complexity motivates **stronger reporting standards**, not by itself a rejection of RL-from-scratch; and
> (iv) reposition SSRL as a **structured-interaction exception regime**, not a generic pretraining recommendation.

---

> > ### Author Rebuttal · Reviewer_wCam · 2026-04-02
> >
> > Thank you for the detailed response. I am happy to increase my score but I have a few more questions/comments:
> > 1. Stronger evidence requirements: I agree, I feel the issue with the particular section there is that it reads as if obtaining supervised signals are always easy. The paper should be clear that it's not even about whether RL is expensive, have "poor quality" (not sure what this means), or exploration is intrinsic to the task (again, not sure what this means); but rather whether one can obtain meaningful supervised signal in the first place.
> > 2. I still find the SSRL section a little bit unclear and potentially sidetracking from the main position. Are the authors intending to say that SSRL would be better than RL in some case, e.g. more scalable through hindsight experience replay/contrastive RL/learning dynamics?
> > 3. Regarding the rewards, there could be a difference between what practitioners see vs what the learners see. Here do we intend the learners to receive auditable/compositional rewards so that they can explicitly reason, say which component is it failing?

---

### Official Review · Reviewer_jiAL · 2026-03-14

**Significance:** 3
**Argument Clarity:** 3
**Rating:** 4
**Confidence:** 3

**Questions:**

Please see "Weakness"

**Alternative Views Section:**

Yes

**Compliance With Llm Reviewing Policy A Conservative:**

Affirmed.

**Discussion Potential:**

3

**Paper Summary:**

This position paper argues that RL should be treated as a post-training adjustment for foundation models rather than for capability creation or early-stage training. The authors say that current training pipelines work best when pre-training and cold-start SFT establish a reasoning structure, after which RL refines behaviors such as correctness, constraint satisfaction, alignment, etc. The author seeks to assess the increasing tendency in LLMs community to treat RL-first strategy as the primary path to reasoning capability in LLMs. Overall, the central issue pointed by author is if RL should be regarded as a high-cost behavioral adjustment operator rather than a way to enable reasoning ability itself.

**Position:**

Yes

**Position In Title:**

Yes

**Related Work:**

3

**Strengths And Weaknesses:**

Strength:

1. The paper directly engages with a major topic in modern LLM training, RLHF v.s. SFT vs RL-first training pipelines. The topic is very timely and important, and surely has great impact on LLMs community. This discussion is very relevant since RL-based post-training is now widely used in foundation model alignment.

2. A central insight of the paper is that RL often encourages probability mass (density) toward behaviors that are already present rather than creating new reasoning abilities. This perspective provides an insightful distinction between capability creation (pre-training and SFT) and behavior refinement (RL).

3. The paper highlights that RL pipelines often hide sampling cost, which an under-discussed issue in current research.

4. The paper also provides actionable recommendations, including explicit entry criteria for applying RL, verifiable rewards, transparent reporting, etc.

Weakness:

1. While the story told by the paper is clear, many researchers already informally view RLHF as a behavior-shaping mechanism rather than capability creation like [1].

2. The cross-domain examples (e.g., R1-style pipelines) are good but described too briefly. More concrete comparisons between staged and RL-first approaches can help improve clarity.

3. I feel some claims that "RL Enhances Rather Than Enables Reasoning" is still too strong. It seems to imply that reasoning ability cannot emerge from RL. For example, DeepSeek R1 paper seems to show that RL can directly enhance the reasoning of LLMs without SFT. How do you the techniques in DeepSeek R1 paper?

4. Are there any domains where RL training can create capabilities, and how would the proposed framework account for them?

[1] Behavior injection: Preparing language models for reinforcement learning. NeurIPS 2025

**Support:**

2

---

> ### Author Rebuttal · Authors · 2026-03-27
>
> We thank the reviewer **jiAL** for the thoughtful and balanced assessment. We are encouraged that the reviewer finds the topic timely, the capability-vs-adjustment distinction insightful, and the practical recommendations useful. The comments help sharpen an important point: the contribution of the paper is **not merely to restate the informal intuition that RL often shapes behavior**, but to turn that intuition into a **more explicit, falsifiable, and compute-accountable framework** with operational entry criteria, reporting standards, and clearly stated exception cases.
>
> On the point that many researchers already informally view RLHF as behavior shaping, we agree that this intuition is already present in the community; our contribution is to make it more explicit, operational, and testable. Our intended contribution is therefore not to claim ownership of that intuition, but to **formalize and operationalize it** into a clearer methodological stance. In particular, the paper does not stop at saying that “RL refines behavior”; it translates this into concrete recommendations around **stage discipline, auditable reward design, lightweight guardrails, and compute-accountable reporting**, together with explicit criteria for when RL-first may be justified. We will revise the manuscript to make this **framework contribution** more explicit, and also strengthen the cross-domain comparison between staged and RL-first pipelines.
>
> On the concern that the phrase “**RL enhances rather than enables reasoning**” may be too strong, we agree that this wording can be read too absolutely. Our intended claim is narrower: **the default interpretation of current evidence should be conservative**. That is, RL is currently more strongly supported as a mechanism for **selection, amplification, and stabilization** of behaviors that are already at least partially elicitable than as a reliable default recipe for creating reasoning from scratch. This is precisely why the paper treats DeepSeek-R1-Zero as a serious alternative view rather than dismissing it. Our point is not to deny the phenomenon, but to question the strongest causal interpretation. In our view, DeepSeek-R1-type results are important **stress tests** for the position, but they do not by themselves establish that RL should be treated as the default capability-creation mechanism: they remain compatible with a selection/amplification account over substantial pretrained structure, and in practice such pipelines also report undesirable behaviors and often reintroduce cold-start data. We will revise the wording to make this narrower claim more explicit and avoid any categorical reading. More broadly, the reviewer’s questions do not weaken the position; they underscore why the field needs to distinguish three separate issues that are often conflated in current discussions: whether RL-induced behaviors are observed, how those behaviors should be causally interpreted, and what training recipe should be treated as the default in practice.
>
> On the question of whether there are domains in which RL may genuinely create capabilities, our framework is intended to **allow such cases rather than rule them out**. We believe RL-first is most defensible when **high-quality supervision is genuinely scarce, verification is strong and low-noise, and interaction is itself the main route by which the target behavior can be discovered**. In such regimes, our argument is not “do not use RL,” but rather “treat RL-first as an exceptional claim that deserves stronger evidence,” including transparent sampling budgets, stability reporting, verifier robustness, and parity-controlled comparisons against minimal cold-start scaffolds. We will revise the paper to make these exception conditions more explicit.
>
> > Concretely, in the revision we will:
> (i) clarify that the paper’s contribution is to **formalize an existing intuition into an operational framework** with falsifiable checks and reporting obligations;
> (ii) expand the **cross-domain staged-vs-RL-first comparison** with more concrete examples;
> (iii) soften any wording that could be read as an impossibility claim, especially around **“enables” vs. “enhances”**; and
> (iv) explicitly state the regimes in which **RL-first may be justified** and what evidence would be required to support such claims.

---

> > ### Author Rebuttal · Reviewer_jiAL · 2026-04-05
> >
> > Author has reached back with detailed explanation to my questions regarding the wording. I'm willing to keep my positive view for now.

---

### Official Review · Reviewer_fy7D · 2026-03-25

**Significance:** 3
**Argument Clarity:** 2
**Rating:** 4
**Confidence:** 3

**Questions:**

Please see weaknesses

**Alternative Views Section:**

Yes

**Compliance With Llm Reviewing Policy A Conservative:**

Affirmed.

**Discussion Potential:**

3

**Paper Summary:**

This position paper argues against the widespread adoption of Reinforcement learning based post training of LLMs with the intention of improving reasoning capabilities of these models.  They argue that ‘RL creates reasoning” is not strongly backed by evidence, although RL has been good for post-training. However, the RL has been shown to reinforce patterns already present in the SFT stage. They further state that their position entails few practical implications like cold-start SFT followed by RL refinement should be the default and the need for higher bar of evidence for claims on reasoning based on RL rather than SFT. Furthermore, they say simple verifiable rewards should be used more than reward engineering. Overall, RL should be used as an refinement of posterior distribution rather than providing probability mass when it is not present in pretraining or SFT.

**Position:**

Yes

**Position In Title:**

Yes

**Related Work:**

2

**Strengths And Weaknesses:**

### Strengths:
1. The paper makes its position very clear from the start and builds evidence supporting its position.
1. The differentiation of the capabilities of various stages of post-training is very pertinent and aptly disambiguated.
1. Some of the action items presented are reasonable and merit attention from the research community.
1. Alternative viewpoints are well discussed

### Weaknesses:
1. The core claim of reasoning via RL training is somewhat not well established. The claim that RL cannot provide probability mass when sft does not already have it needs deeper and rigorous treatment.
1. Although verifiable rewards are better when available, it is simply the case that many a times, these are not available and there is a need for reward engineering.

**Support:**

3

---

> ### Author Rebuttal · Authors · 2026-03-27
>
> We thank the reviewer **fy7D** for engaging directly with the two most important pressure points of our position. These comments help sharpen the precise scope of our claim, rather than weakening the paper’s central position. Our paper is not intended as an impossibility claim that RL can never produce qualitatively new behaviors. Rather, our claim is a **default scientific interpretation and methodological recommendation**: under current evidence, RL is more convincingly supported as a mechanism for **selection, amplification, and stabilization** of behaviors that are already at least partially elicitable, than as a reliable default recipe for creating reasoning from scratch.
>
> On the first point, the reviewer is right that an impossibility-style reading of statements such as “RL cannot provide probability mass when SFT does not already have it” would require a deeper theoretical treatment than a position paper can provide. However, that is not the standard we intend to impose. Our actual claim is narrower and falsifiable: **claims that RL creates reasoning from scratch should carry a higher burden of evidence than claims that RL refines or amplifies latent reasoning behavior.** In our view, this requires parity-controlled comparisons that vary cold-start strength, supervision quality, and RL budget, rather than attributing gains to RL by default. We will revise the wording to make this distinction more explicit and avoid an overly absolute reading.
>
> On the second point, we do not argue that reward engineering is unnecessary. Rather, we argue against treating increasing reward complexity as the **default response to failure**, since more complex proxies can simply move the failure surface rather than remove it. Our recommendation is therefore a **default preference for reward minimalism when auditable signals are available**, and a **higher standard of robustness evaluation when engineered rewards are unavoidable.** We will revise the manuscript to make this boundary condition explicit.
>
> More broadly, the reviewer’s two concerns actually reinforce the need for the paper’s central recommendation: the field needs **stronger discipline in attributing RL gains**, and a clearer separation between **capability creation, capability elicitation, and reward-induced behavior shaping.** We believe this narrower formulation preserves the core position while making it more precise and more useful to the community.
>
> > Concretely, in the revision we will:
> (i) replace any wording that could be read as an impossibility claim with a more precise statement about reliability, controllability, and burden of proof;
> (ii) explicitly distinguish **reasoning creation** from **reasoning elicitation/refinement**; and
> (iii) clarify that when verifiable rewards are unavailable, engineered rewards remain useful but must be treated as fallible proxies and accompanied by stronger robustness and stress-testing evidence.

---

> > ### Author Rebuttal · Reviewer_fy7D · 2026-04-04
> >
> > Thanks for the response. I'll keep my score.

---

### Decision · Program_Chairs · 2026-04-30

**Decision:**

Accept (regular)

**Comment:**

The paper presents a clear and timely position: RL should primarily be viewed as a post-training adjustment mechanism rather than a default approach for capability creation. Reviewers agree that the topic is important and that the distinction between capability creation and refinement is valuable, with useful practical recommendations.

The main concerns relate to framing rather than substance. Several reviewers found some claims overstated and asked for clearer acknowledgment of exception regimes (e.g., RL-first when supervision is scarce), better definition of reward design concepts, and a more focused presentation. One reviewer remained more cautious due to the extent of the implied revisions.

The rebuttal addressed these issues well, clarifying that the claim is methodological rather than absolute, and committing to concrete revisions. Most reviewers considered their concerns resolved or minor.

Overall, I recommend weak acceptance. The paper makes a meaningful position-track contribution and is likely to stimulate discussion, but requires nontrivial revisions to align its framing with the clarified, more precise claim.